# Association of rs738409 Polymorphism in Adiponutrin Gene with Liver Steatosis and Atherosclerosis Risk Factors in Greek Children and Adolescents

**DOI:** 10.3390/nu14173452

**Published:** 2022-08-23

**Authors:** Eleanna Stasinou, Elpida Emmanouilidou-Fotoulaki, Maria Kavga, Fotini Sotiriadou, Alexandros F. Lambropoulos, Maria Fotoulaki, Kyriaki Papadopoulou-Legbelou

**Affiliations:** 14th Department of Pediatrics, “Papageorgiou” General Hospital, 56429 Thessaloniki, Greece; 21st Department of Pediatrics, Ippokrateio General Hospital, 11527 Thessaloniki, Greece; 31st Department of Obstetrics & Gynecology, Faculty of Medicine, “Papageorgiou” General Hospital, Aristotle University of Thessaloniki, 54124 Thessaloniki, Greece

**Keywords:** patatin-like phospholipase domain-containing 3 genotype, non-alcoholic fatty liver disease, non-alcoholic steatohepatitis, dyslipidemia, children

## Abstract

Non-alcoholic fatty liver disease (NAFLD) shares several risk factors with atherosclerosis, as it is associated with components of the metabolic syndrome. However, genetic variations have also been linked to the risk of NAFLD, such as adiponutrin/patatin-like phospholipase domain-containing the protein 3 (PNPLA3) rs738409 polymorphism. The aim of the study was to determine the associations of thePNPLA3 rs738409 polymorphism with NAFLD and atherosclerosis risk factors in children and adolescents from northern Greece. A total of 91 children/adolescents who followed a Mediterranean eating pattern with no particular restrictions were studied. They were divided into three subgroups, according to their body mass index (BMI) and the presence or absence of liver disease. Diagnosis of NAFLD was based on a liver ultrasound, while the distribution of the PNPLA3 rs738409 polymorphism was investigated in all the participants. From the components of metabolic syndrome, only BMI, waist circumference, blood pressure, and the homeostasis model of insulin resistance (HOMA-IR) differed significantly between groups. The rs738409 polymorphism was significantly associated with BMI and NAFLD, while lipid values had no significant association with either NAFLD or gene polymorphism. This study shows that in Greekchildren, there is a significant association between the rs738409polymorphism in the PNPLA3 gene and hepatic steatosis, regardless of bodyweight.

## 1. Introduction

The process of atherosclerosis begins in childhood, and apart from the well-known risk factors, major genetic determinants, such as those influencing the lipid profile, may play a key role in the development of cardiovascular diseases [1,2,3].

On the other hand, non-alcoholic fatty liver disease (NAFLD) is the most common form of chronic liver disease in children, having an increasing prevalence worldwide. The disease shares several risk factors with atherosclerosis, as it is associated with components of the metabolic syndrome (abdominal obesity, insulin resistance, hypertriglyceridemia) [4,5]. The increased risk of atherosclerosis in NAFLD patients has been associated with atherogenic dyslipidemia rather than hepatic steatosis [6]. Patients with features of metabolic syndrome may have atherogenic dyslipidemia, which is characterized by high triglyceride (TG) levels and low high-density lipoprotein-cholesterol (HDL-C) levels [7,8,9].

Furthermore, genetic variations have also been associated with the risk of NAFLD or non-alcoholic steatohepatitis (NASH). Adiponutrin/patatin-like phospholipase domain-containing protein 3 (PNPLA3) or adiponutrin (an adipocyte-specific protein) belongs to a group of lipid-metabolizing enzymes, which has the highest expression in the hepatocytes. The PNPLA3 gene plays a critical role in TG metabolism by mediating a rate-limiting step in TG hydrolysis [10,11]. Adiponutrin has been shown to possess triglyceride hydrolase activity by breaking down TG into fatty acids and glycerol, so that they can be absorbed by the intestinal mucosa [12]. Moreover, adiponutrin participates in a metabolic pathway of TG synthesis, regardless of the action of acetyl-CoA [13,14].

The human PNPLA3 gene is located in the long arm (q) of chromosome 22 at position 13.31 (22q13.31). The rs738409 C > G single-nucleotide polymorphism (SNP) in the PNPLA3 gene encodes an isoleucine to methionine substitution at position 148 (I148M). The variant rs738409 in the PNPLA3 gene is considered to be an important genetic determinant in liver steatosis and steatohepatitis [15,16,17]. More specifically, the I148M mutation results in loss of adiponutrin function, leading to impaired lipid catabolism, increased fatty acid secretion, and liver fat deposition [18,19,20,21]. These alterations appear to occur regardless of the presence of features of the metabolic syndrome, such as insulin resistance and obesity [21,22].

In the present study, we aimed to determine the associations of thePNPLA3 rs738409 polymorphism with NAFL or NASH and atherosclerosis risk factors, such as overweight/obesity, increased blood pressure, and abnormal lipid profile, in children and adolescents from northern Greece.

## 2. Materials and Methods

### 2.1. Study Population and Data Collection

A total of 91 children and adolescents, attending for various reasons the pediatric gastroenterology outpatient unit of the 4th Department of Pediatrics of the Aristotle University of Thessaloniki, enrolled in the study. The inclusion criteria for the participants were: age range between 3 and 16 years old, Greek origin, and absence of kinship between the participants. The exclusion criteria were the presence of metabolic diseases, chronic liver diseases, or renal diseases, as well as the administration of any known drug that induces dyslipidemia.

The patients were divided into three subgroups, according to their BMI and the presence or absence of liver disease. The first group consisted of 31 overweight/obese children with NAFLD or NASH. The second group consisted of 33 overweight/obese children without NAFLD or NASH, and the third group consisted of 27 normal-weight children with NAFLD or NASH. All children with hepatic steatosis were further investigated for other known causes of chronic liver disease, so our sample included only subjects with hepatic steatosis of unknown origin.

### 2.2. Anthropometric and Biochemical Measures

Detailed records of previous medical and family histories were reviewed. All the participants followed a Mediterranean eating pattern, with no particular restrictions. They underwent a systematic physical examination by the same pediatrician, and blood samples were collected for determination of biochemical parameters. Body mass index (BMI) was calculated using the formula body weight (kilograms)/height(meters)^2^. Children with BMI ≥ 85th and <95th percentile for age and gender were considered overweight, while children with BMI ≥ 95thpercentile for age and gender were considered obese [23]. In order to assess abdominal obesity, waist circumference (WC) was measured around the abdomen, in the middle of the distance between the lowest rib and the top of the hipbone, with the participant in the upright position. Values were compared with reference growth charts [24].

Blood pressure (BP) was measured using the appropriate cuff size, with the participant in the sitting position. According to the guidelines for children and adolescences, systolic and diastolic BP > 95th percentile for sex, age, and height for children < 13 years and above 130/80 for teenagers > 13 years of age were considered hypertension [25]. All children with hypertension were thoroughly investigated for secondary causes of hypertension. No cases of secondary hypertension were detected, and therefore, no child was excluded from the study.

Measurements of biochemical parameters, including aspartate aminotransferase (AST), alanine aminotransferase (ALT), gamma-glutamyl transferase(γGT), alkaline phosphatase (ALP), and glucose and total protein levels, as well as lipid profile were obtained in a morning blood sample after overnight fasting using the Abbott Architect c16000 Automatic Biochemistry Analyzer, Abbott Medical Japan.

The lipid profile included measurements of total cholesterol (TC), low-density lipoprotein-cholesterol (LDL-C), HDL-C, and TG. Normal lipid levels were considered: TC values < 200 mg/dL, LDL-C < 130 mg/dL, HDL-C > 40 mg/dL, and TG < 100 mg/dL for children up to 10 years of age and <130 mg/dL in children over 10 years old, according to the Expert Panel Guidelines for dyslipidemia in children [26].

In addition, fasting insulin levels were measured by direct chemiluminescence, using ADVIA Centaur XP immunoassay, Siemens Healthineers, and the homeostasis model assessment of insulin resistance (HOMA-IR) index was used to evaluate insulin resistance.

Diagnosis of NAFLD was based on liver ultrasound. Liver biopsy was performed in 11 patients and confirmed the diagnosis of steatohepatitis (NASH). Selection criteria for liver biopsy were positive family history for NAFL/NASH or diabetes, hepatosplenomegaly in clinical examination, and ALT values > 80 IU/mL [27].

### 2.3. Genotyping

DNA was extracted using the QIA amp DNA Blood Mini Kit (Qiagen, Venlo, Limburg, The Netherlands), and DNA samples were amplified with PCR as previously described, using the sense primer 5′-TGGGCCTGAAGTCCGAGGGT-3′ and antisense primer 5′-CCGACACCAGTGCCCTGCAG-3′ for the rs738409 polymorphism [28,29]. The success rates of these assays were >99.0%. The frequency distribution of PNPLA3 rs738409 SNP was in Hardy–Weinberg equilibrium. The study was performed in accordance with the principles outlined in the 1975 Declaration of Helsinki. Informed consent was obtained from all subjects involved in the study.

### 2.4. Statistical Analysis

Normality was tested both graphically (histograms, Q-Q plots, and boxplots) and with statistical tests (Shapiro–Wilk and Kolmogorov–Smirnov). When results did not correspond, normality was decided based on graphical methods, as both statistical tests have limitations.

Descriptive analysis for all variables of interest was conducted through mean and standard deviation for normally distributed scale variables (age at diagnosis, WC, TC, HDL-C, LDL-C, glucose, ALP, and total protein), median and interquartile range for skewed scale variables (BMI, SBP, DBP, TG, AST, ALT, γGT, insulin, and HOMA-IR), and absolute (N) and relative (%) frequency for categorical variables (systolic and diastolic BP percentiles, polymorphism Rs738409, NAFLD/NASH, and lipid profile as dichotomous variables) in each group.

As all variables were complete, the complete case analysis was used to address one missing value of the variable “Triglycerides”. Thus, missing values can be considered random, and the estimates obtained are not biased.

In the analytical part, chi-square test was used to determine if categorical variables differed across the three groups. One-way ANOVA and Kruskal–Wallis tests (where normality assumption could not hold) were performed to investigate different parameters in each group. Independent sample *t*-test and non-parametric Mann–Whitney U test were performed to compare whether the distribution of parameters was the same for children with and without NAFLD/NASH.

Moreover, multivariate logistic regression was used to determine if rs738409 polymorphism has an impact on the appearance of NAFLD/NASH, adjusting for other parameters. The model was adjusted for a priori chosen plausible confounders (gender and age at diagnosis), and WC, BMI, AST, and ALT were also entered into the adjusted analyses, based on their statistical significance (at the level of 0.05) on the presence of NAFLD/NASH. Preliminary analyses were performed to ensure that there was no violation of the assumptions of multiple logistic regression. The level of statistical significance (alpha) was set at 0.05, and all *p*-values represent two-tailed tests, unless otherwise noted. The analyses were performed with statistical software R 4.1.2.

## 3. Results

### 3.1. Clinical and Biochemical Parameters by Subgroups of Participants

Three out of thirty-one children (9.7%) in group 1 were overweight and 28/31 (90.3%) were obese, while only one child in group 2 was overweight and 32/33 (97%) were obese. Furthermore, BMI and WC values differed significantly between groups (Table 1).

Blood pressure above the 95th percentile included 11/31 (35.5%) children of group 1, 6/33 (18.2%) of group 2, and none of group 3. In addition, median values of systolic BP (SBP) and diastolic BP (DBP) differed significantly between groups (*p* < 0.01 and *p* = 0.02, respectively) (Table 1).

All children from group 2 had normal AST and ALT levels. However, high AST and/or ALT levels were found in 6/31 and 15/31 children in group 1, respectively, and 9/27 and 11/27 children in group 3, respectively. Furthermore, median values of AST, ALT, and γGT differed significantly between groups (Table 1).

Abnormal lipid profile (at least one pathological parameter) included 19/31 children (61.3%) from group 1, 18/33 (54.5%) from group 2, and 13/27 (48.1%) from group 3. However, most children with low HDL-C levels were from groups 1 and 2 of obese children (35.5% and 33.3%, respectively), while the highest TG levels were found in group 1, compared to groups 2 and 3 (36.7% vs. 27.3% and 29.6%) (*p* = 0.71) (Table 2).

Glucose and insulin values did not differ significantly between groups. However, the highest HOMA-IR values were observed in group 1 patients and the lowest values in group 3 patients with lean NAFLD (*p* = 0.08) (Table 1). All other biochemical parameters studied did not differ significantly between groups (Table 1). In addition, the percentages of patients with NAFL or NASH are shown in Table 1.

### 3.2. Distribution of the rs738409 Polymorphism by Group and Association with NAFLD/NASH

Homozygosity in the polymorphism rs738409 of the G allele (GG) was found in 14 out of 64 (21.9%) overweight/obese children (eleven from group 1 and three from group 2) and in 15 out of 27 (55.6%) normal-weight children with NAFL/NASH from group 3 (Table 1).

Heterozygosity in the polymorphism rs738409 (CG) was found in 14 (21.9%) overweight/obese children (nine from group 1 and five from group 2), as well as in nine (33.3%) normal-weight children from group 3 (Table 1).

The Rs738409 genotype differed significantly between groups (*p* < 0.01) (Table 1). Moreover, genotype was significantly associated with both BMI and NAFLD/NASH (Table 3 and Table 4, respectively). The above results show that homozygosity in the polymorphism rs738409 is associated with NAFL/NASH, but the possible confounding effect of BMI (given that it is associated with both NAFLD/NASH and genotype) was further investigated with logistic regression.

### 3.3. Association of Clinical and Biochemical Parameters with PNPLA3 Gene and NAFL/NASH

The only parameters that significantly differed between different genotypes were BMI (*p* < 0.01) and AST and ALT (*p* = 0.06 and *p* = 0.03, respectively) (Table 3). In addition, NAFL or NASH had a significant association with gene polymorphisms, WC, BMI, AST, and ALT (*p* < 0.01) (Table 4). However, lipid values did not have a significant association with either NAFL/NASH or gene polymorphisms (Table 3 and Table 4). The relationship remained insignificant, even when treating lipid profile parameters as dichotomous with cut-off points for normal values, as described above (TC values < 200 mg/dL, LDL-C < 130 mg/dL, HDL-C ≥ 40 mg/dL, TG < 100 mg/dL for children up to 10 years of age, and <130 mg/dL in children over 10 years old).

Finally, in the adjusted multivariate logistic regression, it was revealed that the rs738409 polymorphism was significantly associated with fatty liver, even after controlling for the possible confounding effect of other variables, such as BMI. The carriers of the G allele were more likely to have NAFLD/NASH than the non-carriers, independently of age, sex, BMI, WC, AST, and ALT. More specifically, heterozygosity in the polymorphism rs738409 (CG) increased 6-fold the odds of NAFLD/NASH (OR = 6.04, 95% CI [1.27, 33.63]), while homozygosity in the polymorphism rs738409 of the G allele (GG) was associated with almost a 20-fold increase in the likelihood of developing liver disease (OR = 19.27, 95% CI [2.92, 213.31]), holding all other variables constant (Table 5)

## 4. Discussion

This study confirms that the rs738409 polymorphism in the PNPLA3 gene is associated with NAFDL in a Greek pediatric population. To our knowledge, this is the first study that investigates this association in the Greek population. Furthermore, apart from the well-known association between the rs738409 polymorphism in the PNPLA3 gene and NAFDL, we have also investigated the association of the polymorphism with atherosclerosis risk factors, such as overweight/obesity, abnormal lipid profile, and high blood pressure.

Our study shows that Greek children with hepatic steatosis had a significant percentage of the rs738409 polymorphism in the PNPLA3 gene, regardless of body weight. Although obesity continues to be a major risk factor for hepatic steatosis, genetic background appears to burden both obese and nonobese patients. Other studies also support the lack of a significant association between the rs738409 polymorphism and BMI or other components of metabolic syndrome, such as insulin resistance [4,30,31,32]. However, there are also studies that emphasize the strong influence of obesity and malnutrition in developing NAFLD in adolescence [33,34]. In addition, the frequency of the rs738409 G allele in our study agrees with other studies conducted on children worldwide [4,21,35,36,37].

In our study, the genotypes showed a statistically significant association with BMI and AST/ALT levels. In addition, NAFL or NASH had a significant association with gene polymorphisms, WC, BMI, and AST/ALT, showing that the coexistence of obesity could worsen fat infiltration. The study of Kollerits et al. also showed a significant association between the PNPLA3 gene and higher levels of AST, ALT, and γGT [38]. As the authors report, it is unclear whether the hepatic lipid storage leads to an increase in liver enzymes, or it is associated with subtle hepatic dysfunction (Kollerits). In addition, since ALT is the most specific liver function parameter, it may be an indicator of subclinical liver dysfunction and the result of increased hepatic fat accumulation [38]. Other studies also showed a significant correlation between elevated aminotransferase levels, particularly ALT, and the presence of the polymorphism in both obese and normal-weight children [17,18,39,40,41].

Liver disease has been strongly linked with a high prevalence of type 2 diabetes in adults, while genetic associations between these conditions are still under investigation throughout the world [32]. Indeed, in the past, Speliotes et al. [31] did not manage to demonstrate any correlation between diabetes and the PNPLA3 variant. However, a very recent study with a large sample coming from Taiwan clearly documents that adults with diabetes type 2 and the rs738409 GG genotype are more vulnerable to liver disease [42]. Moreover, in the Chinese adult population, the rs738409 C > G variant not only is not significantly correlated with type 2 diabetes, but also seems to reduce the incidence of the disease [43]. Finally, data from Greece referring only to the adult population support that the PNPLA3 variant plays a vital role in hepatic steatosis development in patients with type 2 diabetes [44].

In our study, no significant association was found between the PNPLA3 gene and the other components of metabolic syndrome studied, such as BP, insulin resistance, and lipid profile. Furthermore, although the percentages of abnormal lipid profile were similar across the three groups of the polymorphisms, obese children with or without NAFL/NASH had lower HDL-C values, while the highest TG values were observed in obese children with NAFL/NASH. As lipid abnormalities are common in obese patients, the coexistence of NAFL/NASH further affects the lipid profile, with a major increase in TG levels. Other studies also showed that the PNPLA3 polymorphism was related to liver fat, but there was no statistically significant correlation with insulin resistance (HOMA-IR) or other components of metabolic syndrome [30,31,32].

On the other hand, NAFLD has been associated with components of metabolic syndrome and consequently with atherosclerosis. There are studies showing an association between abdominal obesity or insulin resistance and endothelial dysfunction [45,46]. Therefore, according to the European Guidelines for the management of NAFLD, it is recommended to measure HOMA-IR in all patients, because this could help identify patients at higher risk of progressive liver disease [47].

Finally, the role of the rs738409 PNPLA3 polymorphism in atherosclerosis development is still uncertain [46,48]. In the study of Petta et al., carotid atherosclerosis (c-IMT) was associated with the GG genotype, especially in young patients with NAFLD. The authors assume that vascular damage is caused by the same pathway that causes liver disease [48]. In addition, in the study of Di Costanzo et al., the rs738409 GG phenotype was associated with increased c-IMT only in patients with metabolic syndrome [49]. As for our study, we did not investigate our patients for subclinical atherosclerosis, as many of our participants were below the age of eight years old, an age at which subclinical atherosclerosis indices are usually normal. However, all other parameters studied, such as obesity, abnormal lipid profile, and hypertension, were not statistically different between groups. On the other hand, Castaldo et al. did not find an association between the rs738409 polymorphism and carotid intima-media thickness, but the rs738409 polymorphism was associated with ALT levels, as was seen in our study [50].

## 5. Limitations

Although the sample of our study was satisfactory for conducting a monocentric study, the participants came from a small region of our country and are not necessarily representative of the Greek population. The generalizability (external validity) of the results, therefore, should be made with caution. Internal validity, however, is not problematic, as the population in the three groups was homogenous, limiting the potential confounding effect and ensuring the comparability of information between groups. Other plausible sources of bias were limited by applying eligibility and exclusion criteria common for all study subgroups. Nevertheless, a multicenter study on different populations is required, in order to draw safe conclusions about the frequency of this particular genotype and its impact on NAFLD/NASH in the Greek population.

## 6. Conclusions

Our study shows that frequencies of the rs738409 PNPLA3 variants were greater in both obese and nonobese pediatric patients of Greek origin with hepatic steatosis. Hence, the contribution of the adiponutrin gene to the natural history of hepatic steatosis is important for the development of future prevention or therapeutic strategies aimed at avoiding the progression of hepatic steatosis to more severe forms of liver disease.

## Figures and Tables

**Table 1 nutrients-14-03452-t001:** Demographic, basic clinical, and laboratory characteristics by group of participants (group 1: overweight/obese with NAFLD or NASH; group 2: overweight/obese without NAFLD/NASH; and group 3: normal-weight children with NAFLD/NASH).

Total Sample (*N* = 91)				
	Overweight/Obese with NAFDL/NASH *N* = 31 (34%)	Overweight/Obese without NAFDL/NASH *N* = 33 (36%)	Normal Weight with NAFDL/NASH *N* = 27 (30%)	
	*N* (%)	*N* (%)	*N* (%)	*χ*^2^ (*p*-Value)
Gender				1.37 (0.50) ##
Male	20 (64.5)	17 (51.5)	14 (51.9)	
Female	11 (35.5)	16 (48.5)	13 (48.1)	
Systolic BP (percentile)				FISHER’s exact test (<0.01) ***
<90	20 (64.5)	24 (72.7)	27 (100)	
90–95	0 (0)	3 (9.1)	0 (0)	
95–99	8 (25.8)	4 (12.1)	0 (0)	
>99	3 (9.7)	2 (6.1)	0 (0)	
Diastolic BP (percentile)				FISHER’s exact test (0.1668) ##
<90	27 (87.1)	30 (90.9)	27 (100)	
90–95	0 (0)	0 (0)	0 (0)	
95–99	4 (12.9)	3 (9.1)	0 (0)	
>99	0 (0)	0 (0)	0 (0)	
Rs738409 polymorphism				27.54 (<0.01) **
CC	11 (35.5)	25 (75.8)	3 (11.1)	
CG	9 (29)	5 (15.2)	9 (33.3)	
GG	11 (35.5)	3 (9.1)	15 (55.6)	
NAFLD	27/31 (87.1)	0	20/27 (74.1)	
NASH	4/31 (12.9)	0	7/27 (25.9)	
	Mean (SD)	Mean (SD)	Mean (SD)	F ^a^ (*p*-value)
Age at diagnosis (years)	10.71 (2.55)	10.18 (2.51)	9.19 (3.79)	1.56 (0.22) ##
Waist circumference (cm)	75.95 (13.94)	77.28 (11.92)	58.17 (8.14)	34.11 (<0.01) ***
Total cholesterol (mg/dL)	168.94 (37.37)	173.76 (32.26)	165.81 (30.09)	0.43 (0.65) ##
HDL-C (mg/dL)	41.10 (11.48)	46.48 (14.00)	47.30 (10.80)	1.96 (0.15) ##
ALP (U/L)	240.77 (94.9)	221.15 (69.73)	229.78 (53.78)	0.44 (0.64) ##
Glucose (mg/dL)	82.55 (9.6)	84.61 (7.16)	83.74 (9.3)	1.96 (0.15) ##
Total protein (g/dL)	7.29 (0.82)	7.22 (0.83)	7.37 (0.79)	0.25 (0.78) ##
LDL-C (mg/dL)	102.10 (31.63)	105.73(27.43)	97.52 (25.21)	0.62 (0.54) ##
	Median (IQR)	Median (IQR)	Median (IQR)	H ^b^ (*p*-value)
BMI (kg/m^2^)	26.4 (4.34)	27.82 (4.15)	15.7 (1.9)	56.78 (<0.01) ***
TG (mg/dL)	95.5 (87.75)	90 (47)	83 (53)	1.05 (0.59) ##
AST (IU/mL)	35 (20)	24 (8)	34 (28)	11.45 (<0.01) ***
ALT (IU/mL)	47 (70)	21 (9)	48 (56.5)	17.21 (<0.01) ***
Insulin (μIU/mL)	6.09 (11.77)	4.9 (8.53)	3.21 (11)	4.69 (0.10) ##
Systolic BP	111 (20.5)	109 (16)	98 (7)	17.76 (<0.01) ***
Diastolic BP	65 (18)	62 (20)	58 (9.5)	7.70 (0.02) *
HOMA-IR	2.58 (2.57)	1.56 (1.81)	1.33 (1.7)	5.06 (0.08) #
γGT	23.13 (9)	17.18 (9)	14.78 (4.5)	9.58 (<0.01) **

*p*-value: 0 ‘***’ 0.001 ‘**’ 0.01 ‘*’ 0.05 ‘#’ 0.1 ‘##’ 1. Percentages are reported for the various classes of categorical variables. *p*-values for ANOVA ^a^ or Kruskal–Wallis test ^b^ depending on the type of variables. BMI: body mass index; BP: blood pressure; AST: aspartate aminotransferase; ALT: alanine aminotransferase; γGT: gamma-glutamyl transferase; ALP: alkaline phosphatase; LDL-C: low-density lipoprotein cholesterol; HDL-C: high-density lipoprotein cholesterol; TG: triglycerides; NAFLD: non-alcoholic fatty liver disease; NASH: non-alcoholic steatohepatitis.

**Table 2 nutrients-14-03452-t002:** Abnormal lipid profile by group of participants (group 1: overweight/obese with NAFLD or NASH; group 2: overweight/obese without NAFLD/NASH; and group 3: normal-weight children with NAFLD/NASH).

	Overweight/Obese with NAFDL/NASH (*Ν* = 31)	Overweight/Obese without NAFDL/NASH(*Ν* = 33)	Normal Weight with NAFDL/NASH(*Ν* = 27)	*p*
↑TC (%)	7/31 (22.6)	6/33 (18.2)	6/27 (22.2)	0.89
↑LDL-C (%)	9/31 (29)	6/33 (18.2)	3/27 (11.11)	0.22
↓HDL-C (%)	11/31 (35.5)	11/33 (33.3)	5/27 (18.5)	0.31
↑TG (%)	11/31 (36.70)	9/33 (27.3)	8/27 (29.6)	0.71

TC: total cholesterol; LDL-C: low-density lipoprotein Cholesterol; HDL-C: high-density lipoprotein cholesterol; TG: triglycerides; NAFLD: non-alcoholic fatty liver disease; NASH: non-alcoholic steatohepatitis.

**Table 3 nutrients-14-03452-t003:** Demographic, basic clinical, and laboratory characteristics by genotype of participants.

Total Sample (*N* = 91)				
	CC *N* = 39 (43%)	CG *N* = 23 (25%)	GG *N* = 29 (32%)	
	*N* (%)	*N* (%)	*N* (%)	χ^2^ (*p*-Value)
Gender				0.36 (0.83) ##
Male	23 (59)	13 (56.5)	15 (51.7)	
Female	16 (41)	10 (43.5)	14 (48.3)	
Systolic BP (percentile)				FISHER’s exact test (0.57) ##
<90	31 (79.5)	18 (78.3)	22 (75.9)	
90–95	1 (2.6)	2 (8.7)	0 (0)	
95–99	5 (12.8)	3 (13)	4 (13.8)	
>99	2 (5.1)	0 (0)	3 (10.3)	
Diastolic BP (percentile)				FISHER’s exact test (0.25) ##
<90	38 (97.4)	21 (91.3)	25 (86.2)	
90–95	0 (0)	0 (0)	0 (0)	
95–99	1 (2.6)	2 (8.7)	4 (13.8)	
>99	0 (0)	0 (0)	0 (0)	
GROUP				27.54 (<0.01) ***
Overweight-obese with NAFDL/NASH	11 (28.2)	9 (39.1)	11 (37.9)	
Overweight-obese without NAFDL/NASH	25 (64.1)	5 (21.7)	3 (10.3)	
Normal weight with NAFDL/NASH	3 (7.7)	9 (39.1)	15 (51.7)	
	Mean (SD)	Mean (SD)	Mean (SD)	F ^a^ (*p*-value)
Age at diagnosis (years)	10.59 (2.75)	10 (3.37)	9.41 (2.95)	1.3 (0.28) ##
WC (cm)	73.89 (12.76)	69.83 (13.19)	68.53 (16.96)	1.29 (0.28) ##
TC (mg/dL)	173.72 (28.46)	168.57 (41.58)	165.38 (32.57)	0.54 (0.59) ##
HDL-C (mg/dL)	45.44 (12.85)	42.39 (10.46)	46.14 (13.46)	0.64 (0.53) ##
ALP (U/L)	219.74 (87.83)	242.78 (73.73)	234.90 (55.38)	0.65 (0.52) ##
Glucose (mg/dL)	84.28 (7.17)	84.52 (10.65)	82.10 (8.80)	0.68 (0.51) ##
Total protein (g/dL)	7.21 (0.83)	7.43 (0.99)	7.28 (0.62)	0.53 (0.59) ##
LDL-C (mg/dL)	104.05 (23.29)	104.83 (37.43)	97.17 (26.16)	0.7 (0.5) ##
	Median (IQR)	Median (IQR)	Median (IQR)	H ^b^ (*p*-value)
BMI (kg/m^2^)	27.6 (24.87)	23.67 (11.30)	16.80 (10.86)	15.04 (<0.01) ***
TG (mg/dL)	97 (64.5)	83 (44.5)	85 (73)	1.99 (0.37) ##
AST (IU/mL)	26 (14)	29 (17)	40 (28)	5.58 (0.06) #
ALT (IU/mL)	21 (15.5)	29 (44)	47 (53)	6.96 (0.03)*
Insulin (μIU/mL)	5.01 (11)	3.63 (7.61)	6.09 (9.49)	3.67 (0.16) ##
SBP	100 (21)	104 (13.5)	101 (14)	0.19 (0.91) ##
DBP	62 (22)	61 (10)	60 (10)	0.58 (0.75) ##
HOMA-IR	1.1 (2.57)	0.8 (1.07)	1.43 (1.71)	3.05 (0.22) ##
γGT	16 (8.5)	13 (5)	16 (7)	2.89 (0.24) ##

*p*-value: 0 ‘***’ 0.001 ‘*’ 0.05 ‘#’ 0.1 ‘##’ 1. Percentages are reported for the various classes of categorical variables. *p*-values for ANOVA ^a^ or Kruskal–Wallis test ^b^ depending on the type of variables. BMI: body mass index; WC: waist circumference; SBP: systolic blood pressure; DBP: diastolic blood pressure; AST: aspartate aminotransferase; ALT: alanine aminotransferase; γGT: gamma-glutamyl transferase; ALP: alkaline phosphatase; TC: total cholesterol; LDL-C: low-density lipoprotein cholesterol; HDL-C: high-density lipoprotein cholesterol; TG: triglycerides.

**Table 4 nutrients-14-03452-t004:** Demographic, basic clinical, and laboratory characteristics in children with and without NAFLD/NASH.

	Without NAFLD/NASH *N* = 33 (36.3%)	With NAFLD/NASH *N* = 58 (63.7%)	
	*N* (%)	*N* (%)	χ^2^ (*p*-Value)
Gender			0.19 (0.66) ##
Male	17 (51.5)	34 (58.6)	
Female	16 (48.5)	24 (41.4)	
Systolic BP (percentile)			FISHER’s exact test (0.15) ##
<90	24 (72.7)	47 (81)	
90–95	3 (9.1)	0 (0)	
95–99	4 (12.1)	8 (13.8)	
>99	2 (6.1)	3 (5.2)	
Diastolic BP (percentile)			FISHER’s exact test (0.70) ##
<90	30 (90.9)	54 (93.1)	
90–95	0 (0)	0 (0)	
95–99	3 (9.1)	4 (6.9)	
>99	0 (0)	0 (0)	
Rs738409 polymorphism			23.6 (<0.01) ***
CC	25 (75.8)	14 (24.1)	
CG	5 (15.2)	18 (31)	
GG	3 (9.1)	26 (44.8)	
	Mean (SD)	Mean (SD)	t ^a^ (*p*-value)
Age at diagnosis (years)	10.18 (2.51)	10 (3.25)	0.07 (0.78) ##
WC (cm)	77.28 (11.92)	67.67 (14.58)	10.37 (<0.01) ***
TC (mg/dL)	173.76 (32.26)	167.48 (33.92)	0.75 (0.39) ##
HDL-C (mg/dL)	46.48 (14)	43.98 (11.5)	0.85 (0.36) ##
ALP (U/L)	221.15 (69.73)	235.66 (78.04)	0.78 (0.38) ##
Glucose (mg/dL)	84.61(7.16)	83.10 (9.4)	0.63 (0.43) ##
Total protein (g/dL)	7.22 (0.83)	7.33 (0.8)	0.33 (0.57) ##
LDL-C (mg/dL)	105.73 (27.43)	99.97 (28.67)	0.87 (0.35) ##
	Median (IQR)	Median (IQR)	U ^b^ (*p*-value)
BMI (kg/m^2^)	27.82 (4.15)	21.70 (11.18)	1471.5 (<0.01) ***
TG (mg/dL)	90 (47)	87 (67)	981 (0.74) ##
AST (IU/mL)	24.0 (8)	34.5 (24.75)	547.5 (<0.01) ***
ALT (IU/mL)	21 (9)	47.5 (64.75)	482.5 (<0.01) ***
Insulin (μIU/mL)	4.9 (8.53)	5.63 (11.15)	869.5 (0.47) ##
SBP	109 (16)	100 (14)	1022 (0.46) ##
DBP	62 (20)	60 (11.75)	978 (0.71) ##
HOMA-IR	1 (1.81)	1.25 (2.05)	877 (0.51) ##
γGT	16 (9)	15 (7.75)	992.5 (0.77) ##

*p*-value: 0 ‘***’ 0.001 ‘##’ 1. Percentages are reported for the various classes of categorical variables. *p*-values for independent sample *t* test ^a^ or Mann–Whitney U test ^b^ depending on the type of variables. BMI: body mass index; WC: waist circumference; SBP: systolic blood pressure; DBP: diastolic blood pressure; AST: aspartate aminotransferase; ALT: alanine aminotransferase; γGT: gamma-glutamyl transferase; ALP: alkaline phosphatase; TC: total cholesterol; LDL-C: low-density lipoprotein cholesterol; HDL-C: high-density lipoprotein cholesterol; TG: triglycerides, NAFLD: non-alcoholic fatty liver disease; NASH: non-alcoholic steatohepatitis.

**Table 5 nutrients-14-03452-t005:** Adjusted multivariate logistic regression: polymorphism on NAFLD/NASH.

	OR	95% CI	*p*-Value
Rs738409 polymorphismCG	6.04	1.27, 33.63	0.029 *
Rs738409 polymorphismGG	19.27	2.92, 213.31	<0.01 **
Gender, Female	0.66	0.17, 2.50	0.54 ##
Age (years)	1.37	0.96, 2.09	0.10 ##
Waist circumference (cm)	0.94	0.84, 1.03	0.22 ##
BMI (kg/m^2^)	0.87	0.71, 1.06	0.19 ##
AST (IU/mL)	1.03	0.95, 1.12	0.48 ##
ALT (IU/mL)	1.06	1.02, 1.14	0.02 *

*p*-value: 0 ‘**’ 0.01 ‘*’ 0.05 ‘##’ 1. BMI; body mass index; AST: aspartate aminotransferase; ALT: alanine aminotransferase; NAFLD: non-alcoholic fatty liver disease; NASH: non-alcoholic steatohepatitis.

## Data Availability

Data sharing is not applicable to this article.

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
