# Peer review of "Association of rs738409 Polymorphism in Adiponutrin Gene with Liver Steatosis and Atherosclerosis Risk Factors in Greek Children and Adolescents"

_nutrients, 2022, doi:10.3390/nu14173452_

Round 1
Reviewer 1 Report
The Authors investigated the association of adiponutrin gene polymorhism with liver steatosis and atherosclerosis risk factors in a population of 91 Greek children/adolescents.
The study design and analysis are clearly described. Data presentation is clear. The study rationale remains unclear, however, since the results seem to repeat conclusions from ref. 18-20 published over a decade ago. Information on how this study actually contributes to the current state of the art is necessary.
The analysis includes arterial hypertension as one of the risk factors of atheroslerosis. However, hypertension is majorly secondary in the paediatric population. Was diagnostics of hypertension completed in the investigated cohort to ascertain the etiology in the hypertensive subjects?
Also, please explain what was the etiology of liver steatosis in the included subjects if chronic liver disease was an exclusion criterion.
Author Response
Dear Reviewer,
Thank you very much for the time you devoted to our work, as well as for your helpful contribution.
We are now pleased to submit our revision. We have tried to answer all issues.
Please find attached.
We are looking forward to your final decision!
Yours sincerely,
Kyriaki Papadopoulou-Legbelou,MD,PhD
Associate Professor in Paediatrics-Paediatric Cardiology.
Fourth Department of Paediatrics, Medical School, Aristotle University of Thessaloniki ‘Papageorgiou’ General Hospital, Ring Road Nea Efkarpia, Thessaloniki 56403, Greece,
Tel :+302310991463
Fax : +302313323918
Email : kelipap@gmail.com,kpapadopoulou@auth.gr

Reviewer 2 Report
The present manuscript authored by Stasinou et al. reports on the association between rs738409 polymorphism in adiponutrin/patatin-like phospholipase domain-containing protein 3 (PNPLA3) gene and hepatic steatosis independent of body weight in children and adolescents from northern Greece. Although these observations deserve attention, the study needs clarifying on several issues:
- The novelty of this work is hampered by the fact that rs738409 polymorphism in the PNPLA3 gene is a critical genetic factor that confers high-risk to NAFLD is a well-known concept (PMIDs: 25641744, 22719876, 25069572, 34147109, and 34106646) even in children (PMIDs: 32811452, 31216264). Authors should emphasize how they study advances the field.
- How was normality of data tested?
- Was correction for multiple testing taken into account?
- Association between PNPLA3 rs738409 variant and diabetes is highly recommended.
- Authors found that the relationship between rs738409 polymorphism in the PNPLA3 gene and hepatic steatosis is independent of body weight in contrast to other reports in the literature (PMIDs: 28619255, 32561908). This discrepancy should be tackled.
- This work does not provide an in-depth analysis of the association between genetic variant, NAFLD, and atherosclerosis. Indeed, it was recently published that rs738409 polymorphisms do not relate to distinct measures of subclinical atherosclerosis but associate to ALT levels (PMID: 33105679).
Minor comments
- Group 1 / 2 / 3 in Tables should be changed to a more meaningful name (e.g.: obese NAFLD, lean NAFLD,…).
Author Response

(The authors gave the same response as above.)

Round 2
Reviewer 2 Report
Authors have adequately responded to my comments and followed most of my suggestions. The manuscript in its present form is suitable for publication in Nutrients.